# Swin-Transformer-YOLOv5 for Real-Time Wine Grape Bunch Detection

Shenglian Lu [1,2], Xiaoyu Liu [1], Zixuan He [2,3], Xin Zhang [4,*], Wenbo Liu [4,5] and Manoj Karkee [2,3]

1. Guangxi Key Lab of Multisource Information Mining and Security, College of Computer Science and Engineering, Guangxi Normal University, Guilin 541004, China
2. Center for Precision and Automated Agricultural Systems, Washington State University, Prosser, WA 99350, USA
3. Department of Biological Systems Engineering, Washington State University, Pullman, WA 99164, USA
4. Department of Agricultural and Biological Engineering, Mississippi State University, Mississippi State, MS 39762, USA
5. Coastal Research and Extension Center, Mississippi State University, Pascagoula, MS 39567, USA
* Correspondence: xzhang@abe.msstate.edu; Tel.: +1-662-325-1983

**Abstract:** Precise canopy management is critical in vineyards for premium wine production because maximum crop load does not guarantee the best economic return for wine producers. The growers keep track of the number of grape bunches during the entire growing season for optimizing crop load per vine. Manual counting of grape bunches can be highly labor-intensive and error prone. Thus, an integrated, novel detection model, Swin-transformer-YOLOv5, was proposed for real-time wine grape bunch detection. The research was conducted on two varieties of Chardonnay and Merlot from July to September 2019. The performance of Swin-T-YOLOv5 was compared against commonly used detectors. All models were comprehensively tested under different conditions, including two weather conditions, two berry maturity stages, and three sunlight intensities. The proposed Swin-T-YOLOv5 outperformed others for grape bunch detection, with mean average precision (mAP) of up to 97% and F1-score of 0.89 on cloudy days. This mAP was ~44%, 18%, 14%, and 4% greater than Faster R-CNN, YOLOv3, YOLOv4, and YOLOv5, respectively. Swin-T-YOLOv5 achieved an $R^2$ of 0.91 and RMSE of 2.4 (number of grape bunches) compared with the ground truth on Chardonnay. Swin-T-YOLOv5 can serve as a reliable digital tool to help growers perform precision canopy management in vineyards.

**Keywords:** computer vision; crop load estimation; deep learning; full growth season; in-field imaging; object detection; precision viticulture; vineyard management; wine grape; yolo

## 1. Introduction

The overall grape production in the United States has reached 6.05 million tons in 2021, in which approximately 5.78 million tons (~96%) were from wine grape production in California and Washington [1]. To maintain the premium quality of wine, vineyards need to be elaborately managed so that the quantity and quality of the grapes can be well balanced for maximum vineyard profitability. Such vineyard management can be difficult because the number of berry bunches should be closely monitored by laborers throughout the entire growing season to avoid a high volume of bunches overburdening the plant and thus the berry composition may not be optimal [2]. Presenting this information can help the growers to timely prune and thin the grape clusters during the growing season. This presents significant challenges for wine grape growers and managers due to the agricultural workforce shrinking and cost increasing. Potentially, this issue might be mitigated by leveraging the superiority of state-of-the-art computer vision technologies and data-driven artificial intelligence (AI) techniques [3].

Object detection is one of the fundamental tasks in computer vision, which is used for detecting instances of one or more classes of objects in digital images. Several common

challenges, that prevent a target object from being successfully detected, include but are not limited to variable outdoor light conditions, scale changes in the objects, small objects, and partially occluded objects. In recent years, numerous deep learning-driven object detectors have been developed for various real-world tasks, such as fully connected networks (FCNs), convolutional neural networks (CNNs), and Vision Transformer. Among these, CNN-based object detectors have demonstrated promising results [4,5]. Generally, CNN-based object detectors can be divided into two types, including one-stage detectors and two-stage detectors. The one-stage detector uses a single network to predict the bounding boxes and calculates the class probabilities of the boxes. Two-stage detector first proposes a set of regions of interest (i.e., region proposal) where the potential bounding box candidates can be infinite, then a classifier processes the region candidates only. Taking a few examples, one-stage detectors include Single Shot Multibox Detector (SSD) [6], RetinaNet [7], Fully Convolutional One-Stage (FCOS) [8], DEtection TRansformer (DETR) [9], EfficientDet [10], You Only Look Once (YOLO) family [11–14], while two-stage detectors include region-based CNN (R-CNN) [15], Fast/Faster R-CNN [16,17], Spatial Pyramid Pooling Networks (SPPNet) [18], Feature Pyramid Network (FPN) [19], and CenterNet2 [20].

As agriculture is being digitalized, both one-stage and two-stage object detectors have been widely applied to various orchard and vineyard scenarios, such as fruit detection and localization, with promising results achieved. Some of the major reasons, which made object detection challenging in agricultural environments, include severe occlusions from non-target objects (e.g., leaves, branches, trellis-wires, and densely clustered fruits) to target objects (e.g., fruit) [21]. Thus, in some cases, the two-stage detectors were preferred by the researchers due to their greater accuracy and robustness. Tu et al. [22] developed an improved model based on multi-scale Faster R-CNN (MS-FRCNN) that used both RGB (i.e., red, green, and blue) and depth images to detect passion fruit. Results indicated that the precision of the proposed MS-FRCNN was improved from 0.85 to 0.93 (by ~10%) compared to generic Faster R-CNN. Gao et al. [21] proposed a Faster R-CNN-based multi-class apple detection model for dense fruit-wall trees. It could detect apples under different canopy conditions, including non-occluded, leaf-occluded, branch/trellis-wire occluded, and fruit-occluded apple fruits, with an average detection accuracy of 0.879 across the four occlusion conditions. Additionally, the model processed each image in 241 ms on average. Although two-stage detectors have shown robustness and promising detection results in agricultural applications, there is still one major concern, the high requirement of computational resources (leading to slow inference speed), to further implement them in the field. Therefore, it has become more popular nowadays to utilize one-stage detectors in identifying objects in orchards and vineyards, particularly using YOLO family models with their feature of real-time detection.

Huang et al. [23] proposed an improved YOLOv3 model for detecting immature apples in orchards, using Cross Stage Partial (CSP)-Darknet53 as the backbone network of the model to improve the detection accuracy. Results showed that the F1-Score and mean Average Precision (mAP) were 0.65 and 0.68, respectively, for those severely occluded fruits. Furthermore, Chen et al. [24] also improved the YOLOv3 model for cherry tomato detection, which adopted a dual-path network [25] to extract features. The model established four feature layers at different scales for multi-scale detection, achieving an overall detection accuracy of 94.3%, recall of 94.1%, F1-Score of 94.2%, and inference speed of 58 ms per image. Lu et al. [26] introduced a convolutional block attention module (CBAM) [27] and embedded a larger-scale feature map to the original YOLOv4 to enhance the detection performance on canopy apples in different growth stages. In general, object detectors tend to have false detections when occlusion occurs, no matter using a one-stage or two-stage detector. YOLO family detectors, like many other widely adopted detectors, could also have information loss affected by canopy occlusions. During the past two years, Vision Transformer has demonstrated outstanding performances in numerous computer vision tasks [28] and, therefore, is worth being further investigated to be employed together with YOLO models in addressing the challenges.

A typical Vision Transformer architecture is based on a self-attention mechanism that can learn the relationships between components of a sequence [28,29]. Among all types, Swin-transformer is a novel backbone network of hierarchical Vision Transformer, using a multi-head self-attention mechanism that can focus on a sequence of image patches to encode global, local, and contextual cues with certain flexibilities [30]. Swin-transformer has already shown its compelling records in various computer vision tasks, including region-level object detection [31], pixel-level semantic segmentation [32], and image-level classification [33]. Particularly, it exhibited strong robustness to severe occlusions from foreground objects, random patch locations, and non-salient background regions. However, using Swin-transformer alone in object detection requires large computing resources as the encoding–decoding structure of the Swin-transformer is different from the conventional CNNs. For example, each encoder of Swin-transformer contains two sublayers. The first sublayer is a multi-head attention layer, and the second sublayer is a fully connected layer, where the residual connections are used between the two sublayers. It can explore the potential of feature representation through a self-attention mechanism [34,35]. Previous studies on public datasets (e.g., COCO [36]) have demonstrated that Swin-transformer outperformed other models on severely occluded objects [37]. Recently, Swin-transformer has also been applied in the agricultural field. For example, Wang et al. [38] proposed "SwinGD" for grape bunch detection using Swin-transformer and Detection Transformer (DETR) models. Results showed that SwinGD achieved 94% of mAP, which was more accurate and robust in overexposed, darkened, and occluded field conditions. Zheng et al. [39] researched a method for the recognition of strawberry appearance quality based on Swin-transformer and Multilayer Perceptron (MLP), or "Swin-MLP", in which Swin-transformer was used to extract strawberry features and MLP was used to identify strawberry according to the imported features. Wang et al. [40] improved the backbone of Swin-transformer and then applied it to identify cucumber leaf diseases using an augmented dataset. The improved model had a strong ability to recognize the diseases with a 99.0% accuracy.

Although many models for fruit detection have been studied in orchards and vineyards [26,41–47], the critical challenges in grape detection in the field environment (e.g., multi-variety, multi-stage of growth, multi-condition of light source) have not yet been fully studied using a combined model of YOLOv5 and Swin-transformer. In this research, to achieve better accuracy and efficiency of grape bunch detection under dense foliage and occlusion conditions in vineyards, we architecturally combined the state-of-the-art, one-stage detector of YOLOv5 and Swin-transformer (i.e., Swin-Transformer-YOLOv5 or Swin-T-YOLOv5), so that the proposed new network structure had the potential to inherently preserve the advantages from both models. The overarching goal of this research was to detect wine grape bunches accurately and efficiently under a complex vineyard environment using the developed Swin-T-YOLOv5. The specific research objectives were to:

- Effectively detect the in-field wine grape bunches by proposing a novel combined network architecture of Swin-T-YOLOv5 using YOLOv5 and Swin-transformer;
- Compare the performance of the developed Swin-T-YOLOv5 with other widely used object detectors, including Faster R-CNN, generic YOLOv3, YOLOv4, and YOLOv5, and investigate the results under different scenarios, including different wine grape varieties (i.e., white variety of Chardonnay and red variety of Merlot), weather conditions (i.e., sunny and cloudy), berry maturities or growth stages (i.e., immature and mature), and sunlight directions/intensities (i.e., morning, noon, and afternoon) in vineyards.

## 2. Materials and Methods

### 2.1. Data Acquisition and Preprocessing

2.1.1. Wine Grape Dataset

The data acquisition and research activities in this study were carried out in a wine vineyard located in Washington State University (WSU) Roza Experimental Orchards,

Prosser, WA. Two different wine grape varieties were selected as the target crops due to their distinct color of berry skin when mature, including Chardonnay (white berries; Figure 1a) and Merlot (red berries; Figure 1d). The color of berry skin for Chardonnay was consistently white throughout the growth season (Figure 1b,c), while the color for Merlot changed from white to red (Figure 1e,f) during the season. There were approximately 10–33 and 12–32 grape bunches per vine for the experimental Chardonnay and Merlot plants in this study. The wine vineyard was maintained by a professional manager for optimal productivity. The row and inter-plant spaces were about 2.5 m and 1.8 m for both varieties.

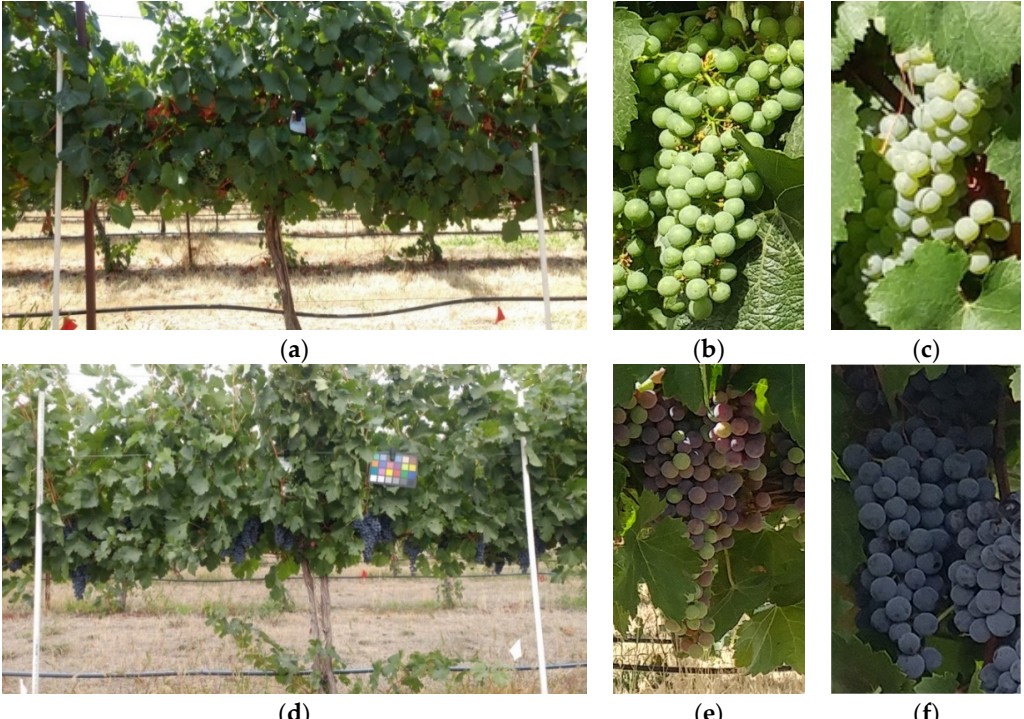

(a)          (b)          (c)

(d)          (e)          (f)

**Figure 1.** Grape dataset acquisition on (**a**) Chardonnay (white color of berry skin when mature; 10–33 grape bunches per plant) with the close-up views of grape bunch during (**b**) immature and (**c**) mature stages, and (**d**) Merlot (red color of berry skin when mature; 12–32 grape bunches per plant) with the close-up views of grape bunch during (**e**) immature and (**f**) mature stages.

The imagery data collection was completed using a Samsung Galaxy S6 smartphone (Samsung Electronics Co., Ltd., Suwon, Republic of Korea) at distances of 1–2 m, while the camera was facing perpendicularly to the canopy. The data collection was carried out during the entire growth season (i.e., from the berries were developed to mature) at a periodical frequency of one day per week and three times per day from 4 July 2019 to 30 September 2019. More specific details were given in Table 1 that the images of the canopies were captured under two weather conditions (i.e., sunny and cloudy), two berry maturity conditions (i.e., immature from 4 July 2019–27 July 2019 and mature from 7 September 2019–30 September 2019), and three sunlight direction/intensity conditions (i.e., morning at 8am–9am, noon at 11am–12pm, and afternoon at 4pm–5pm, Pacific Daylight Time). All these various outdoor conditions largely represented the diversity of the imagery dataset. Note that all images were always acquired from the same side of the canopy. As a result, 459 raw images were collected in total for Chardonnay (234 images) and Merlot (225 images) grape varieties in the original resolution of 5312 × 2988 pixels (Table 2). The specific number of raw images under individual conditions can also be found in Table 1.

**Table 1.** Grape dataset collected under different weather, berry maturity, and sunlight direction/intensity conditions.

| Grape Variety | Weather/Plant/Light Condition | | Number of Raw Images | Total |
|---|---|---|---|---|
| Chardonnay | Weather | Sunny | 169 [5] | 234 |
| | | Cloudy | 65 | |
| | Berry maturity | Immature (white) | 83 | 155 [6] |
| | | Mature (white) | 72 | |
| | Sunlight direction [1] | Morning [2] | 91 | 234 |
| | | Noon [3] | 75 | |
| | | Afternoon [4] | 68 | |
| Merlot | Weather | Sunny | 153 | 225 |
| | | Cloudy | 72 | |
| | Berry maturity | Immature (white/white–red mix) | 81 | 162 [6] |
| | | Mature (red) | 81 | |
| | Sunlight direction | Morning | 82 | 225 |
| | | Noon | 70 | |
| | | Afternoon | 73 | |

[1] All images in this study were taken from the consistent side of the canopy. [2] Morning at 8am–9am (in the direction of the light). [3] Noon at 11am–12pm (maximum solar elevation angle). [4] Afternoon at 4pm–5pm (against the direction of the light). [5] Both sunny and cloudy images were included in the train set. [6] All images were periodically collected from 4 July 2019–30 September 2019, in which the dates were divided into three growth stages, including early stage (4 July 2019–27 July 2019), middle stage (2 August 2019–24 August 2019), and late (harvest) stage (7 September 2019–30 September 2019). During the middle stage, the change in shape and color of the grapes was inconsiderable. Therefore, we only compared the early stage (immature) and late stage (mature).

**Table 2.** Grape imagery dataset and augmentation in this study.

| Dataset | Variety | Color of Berry Skin | Original Image Size (Pixels) | Number of Raw Images | Number of Total Images (After Augmentation) |
|---|---|---|---|---|---|
| Grape | Chardonnay | White | 5312 × 2988 | 234 | 2263 |
| | Merlot | Red | | 225 | 2155 |
| | | Total | | 459 | 4418 |

2.1.2. Dataset Annotation and Augmentation

The raw imagery dataset was manually annotated using the annotation tool of LabelImg [48]. The position of the grape bunch was individually selected using bounding boxes. Clustered grape bunches were also carefully separated. In addition, the "debar" approach was adopted based on our previous publication [26] to separate individual canopies for evaluation purposes only. Once all raw images (Figure 2a) were annotated with manual labels, the dataset was further enriched (Figure 2b–e) by using data enhancement and augmentation library of Imgaug [49]. During data augmentation, the annotated "key points" and "bounding boxes" were transformed accordingly. The enriched dataset can better represent the field conditions of the grape bunches. After augmentation, a dataset containing 4418 images was developed, where a detailed description of the augmented dataset can be found in Table 2. The finalized dataset was further divided into train (80%), validation (10%), and test sets (10%), respectively, for development of grape bunch detection models. Finally, the in-field manual counting of grape bunches was completed during the harvest season on 1 October 2019 after the last dataset was acquired.

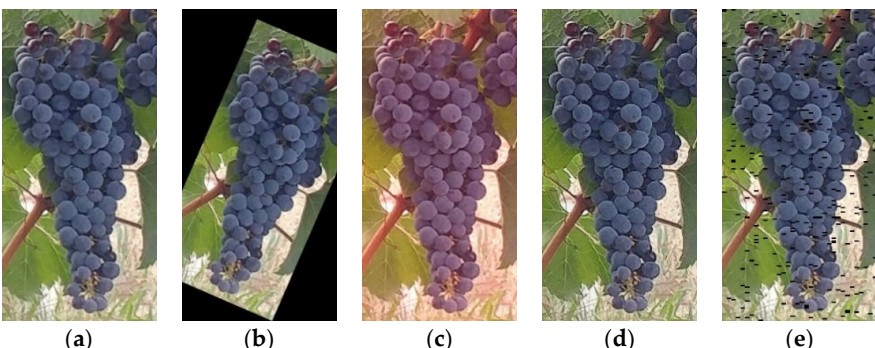

**Figure 2.** Illustrations of the dataset augmentation: (**a**) original image; (**b**) rotation; (**c**) channel enhancement; (**d**) Gaussian blur/noise; and (**e**) rectangle pixel discard.

### 2.2. Grape Bunch Detection Network

### 2.2.1. Swin-Transformer

First, the Swin-transformer architecture [30] was introduced in Figure 3a. It can split the input RGB image into non-overlapping, small patches through a patch partition module. Each patch was treated as a "token" whose features were set as the concatenation of the raw pixel values in the RGB image (i.e., 3 channels). In this study, a patch size of $4 \times 4$ was used and, therefore, the feature dimension per patch was $4 \times 4 \times 3 = 48$. A linear embedding layer was then applied to this raw value feature to project it to an arbitrary dimension (denoted as C in Figure 3a). Swin-transformer was built through replacing the standard multi-head self-attention (MSA) module in a regular Transformer block by an MSA module based on "windows" (i.e., W-MSA) and "shifted windows" (i.e., SW-MSA), while other layers kept the same (Figure 3b). This module was followed by a 2-layer multi-layer perceptron (MLP) with nonlinearity of rectified linear unit (ReLU) in between. A normalization layer (LayerNorm) and a residual connection were applied before and after each MSA module and MLP layer.

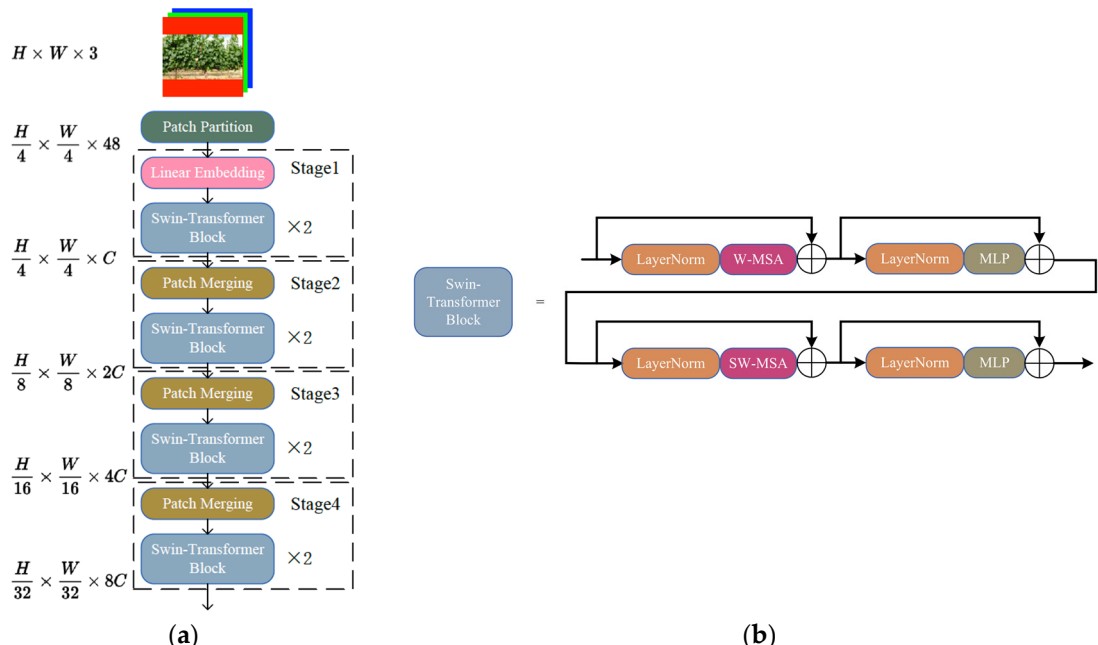

**Figure 3.** (**a**) Overall architecture of Swin-transformer (H and W refer to height and width of the images; C refers to the number of feature channels) and (**b**) two successive Swin-transformer blocks (W-MSA and SW-MSA refer to Window-based multi-head self-attention and shifted window-based MSA; MLP refers to multi-layer perceptron).

After linear projection and reshape operation, a feature map $X \in R^{H \times W \times C}$ becomes $Q, K, V \in R^{N \times C'}$ to provide self-attention, where $N = H \times W$. The output of self-attention is expressed in Equations (1) and (2):

$$\text{Attention} = TV \tag{1}$$

$$T = \text{SoftMax}\left(\frac{QK^T}{\sqrt{d}} + B\right) \tag{2}$$

where $T \in R^{N \times N}$ is the attention matrix representing the relationship between all elements on the feature map and other elements. The output Attention aggregates the global information. The absolute position encoding is to add a learnable parameter to each token before the self-attention calculation, and the relative position encoding is to add a learnable relative position parameter in the calculation process of self-attention. The relative position offset is $B \in \mathbb{R}^{M^2 \times M^2}$, and the value range of each axis is $[-M + 1, M - 1]$. Then W-MSA module was used to reduce the amount of computation. MSA performs the self-attention calculation among all pixels in Vision Transformer (Equation (3)), while W-MSA only performs the self-attention calculation among pixels in the same window (size of $7 \times 7$). With the local window size of $m \times m$, the computational complexity ($\Omega$) of a feature map $X \in R^{H \times W \times C}$ was calculated in Equations (3) and (4):

$$\Omega(\text{MSA}) = 4hwC^2 + 2(hw)^2C \tag{3}$$

$$\Omega(W - \text{MSA}) = 4hwC^2 + 2M^2hwC \tag{4}$$

### 2.2.2. YOLOv5

YOLOv5 (specifically, YOLOv5s) is a recent detection model in YOLO family [13], which has fast inference (detection) speed. In addition, YOLOv5s is a lightweight model with fewer model parameters, which is approximately 10% of the generic YOLOv4, indicating that this model might be more suitable for deployment on embedded devices for real-time object detection. Combined with all these advantages, this study attempted to detect grape bunches in dense canopies using the improved YOLOv5.

In general, YOLOv5 framework includes three parts: backbone, neck, and detection (or output) networks (Figure 4a). The backbone network was used to extract feature maps from the input images with multiple convolutions and merging. A three-layer feature map was then generated in the backbone network in the sizes of $80 \times 80$, $40 \times 40$, and $20 \times 20$ (Figure 4a; left). After backbone network, the neck network contained a series of feature fusion layers that can mix and combine image features. All feature maps in different sizes generated by the backbone network were fused to obtain more context information and reduce the information loss. The characteristic pyramid structure of Feature Pyramid Network (FPN) and Path Aggregation Network (PANet) were adopted during the merging process, where strong semantic features were transferred from top to bottom feature maps using FPN structure. Meanwhile, strong localization features were transferred from lower to higher feature maps using PANet. Overall, the ability of feature fusion in the neck network was enhanced by using FPN and PANet together (Figure 4a; middle). Finally, the detection network was used to give the detection results. It consisted of three detection layers, with the corresponding output feature maps of $80 \times 80$, $40 \times 40$, and $20 \times 20$, which was used to detect objects in the input images. Each detection layer ultimately can output a 21-channel vector and then generate and mark the predicted bounding box and category of the target in the original input images for final detections (Figure 4a; right).

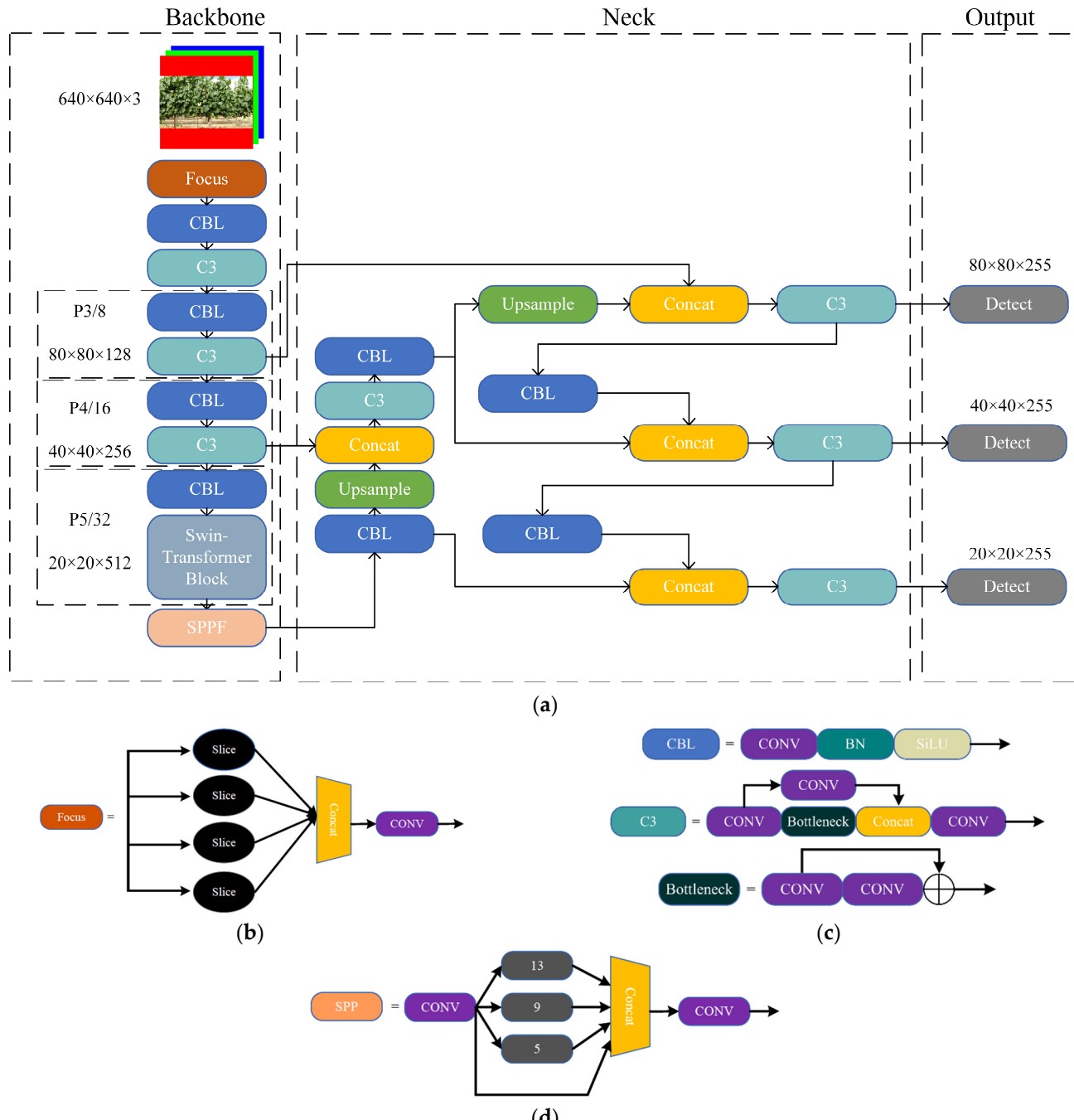

**Figure 4.** (**a**) Integrated architecture of Swin-transformer-YOLOv5; layers of (**b**) focus; (**c**) CBL and cross-stage partial (CSP) bottleneck with 3 convolutions (C3); and (**d**) spatial pyramid pooling (SPP), where CONV, Concat, BN, SiLU, and ADD (⊕) refer to convolutional, concatenate, batch normalization, activation function of sigmoid linear unit, and feature fusion with the number of channels unchanged. P refers to the specific layer of feature map.

Moreover, the focus module of YOLOv5 can slice and concatenate images (Figure 4b), which was designed to reduce the computational load of the model and speed up the training process. It can first split the input 3-channel image into four slices using the slice operation. The four slices were concatenated using the Concat operation, and a convolutional layer (CONV) was then used to generate the output feature map. Figure 4c gave the explanations on some layers/modules in backbone network, including CBL and C3, in which CBL was a standard convolutional module consisting of CONV, batch

normalization (BN), and activation function of sigmoid linear unit (SiLU); C3 was cross stage partial (CSP) bottleneck with 3 CONVs. The initial input was split into two branches, and thus the number of channels of the feature maps was halved by CONV operation in each branch. The output feature maps of the two branches were again connected through the Concat operation. The final output feature map of C3 was generated by CONV. C3 was used to improve inference (test) speed by reducing the size of the model while maintaining desired performances in extracting useful features from images. Finally, spatial pyramid pooling (SPP) module was used to improve the receptive field by converting feature maps of arbitrary size into feature vectors of fixed size (Figure 4d). The feature map was first output through a CONV layer with the kernel size of $1 \times 1$. It was then connected with the output feature map subsampled by three parallel max pooling layers, followed by a CONV layer to output the final feature map.

### 2.2.3. Integration of Swin-Transformer and YOLOv5

To take advantage of both Swin-transformer and YOLOv5, two models were integrated (i.e., Swin-transformer-YOLOv5 or Swin-T-YOLOv5) by replacing the last C3 layer (i.e., with CSP bottleneck and three CONVs) in the backbone network of the original YOLOv5 with Swin-transformer encoder blocks (Figure 4a). Because the resolution of feature maps was $20 \times 20$ at the end of the backbone network, applying Swin-transformer on low-resolution feature maps can reduce computational load and save memory space [50]. Swin-transformer can be used to capture long-distance dependencies and retain different local information [30]. Although such integration may slightly slow down the inference speed of YOLOv5, the detection accuracy can be enhanced. Therefore, our proposed scheme combined YOLOv5s and Swin-transformer, so that the new structure can inherit their advantages and preserve both global and local features. Furthermore, the self-attention mechanism was used to improve the detection accuracy of the integrated model. This integration might be particularly useful for the occluded grape bunches in dense foliage vineyard canopies. Pre-trained YOLOv5s using COCO dataset was adopted during training to improve the generalization ability of the proposed network. In the training phase, we used a partially pre-trained model of YOLOv5s. There were many weights that can be directly transferred from YOLOv5s to Swin-T-YOLOv5 because they shared most of the backbone weights (i.e., blocks 0–7 in Figure 4a "Backbone"), which can save the training time. Then, we added Swin-transformer to the 8th layer of the backbone network (i.e., block 8 in Figure 4a "Backbone"), where this layer and the subsequent nodes were retrained without using the pretrained weights.

The training, validation, and testing steps were carried out on a workstation with an Intel® Xeon® Silver 4114 CPU, 64 GB RAM, NVIDIA RTX3090 GPU (24 GB VRAM), and Ubuntu 20.04 LTS Operating System. Python was used to write program code and call required libraries, such as CUDA, cuDNN, and OpenCV, on top of PyTorch 1.8.1 framework. To comprehensively evaluate the performances, our proposed Swin-T-YOLOv5 was compared against Faster R-CNN [17], YOLOv3 [12], YOLOv4 [11], and YOLOv5 [13], where the training hyperparameters of each model were shown in Table 3. Specific hyperparameters were chosen to generally maintain their generic settings and to meet the hardware requirement of the running platform.

### 2.2.4. Evaluation Metrics

The performance of each model was evaluated using its precision (P), recall (R), F1-score, mean average precision (mAP) (Equations (5)–(8)), and inference (detection) speed per image, in which mAP served as a key metric to assess the overall performance of a model:

$$P = \frac{TP}{TP + FP} \tag{5}$$

$$R = \frac{TP}{TP + FN} \tag{6}$$

$$F_1 = \frac{2 \times P \times R}{P + R} \tag{7}$$

$$mAP = \frac{1}{n} \sum_{k=1}^{k=n} AP_k \tag{8}$$

where true positives (TP) represent the positive samples correctly predicted by the model, true negatives (TN) represent the negative samples correctly predicted by the model, false positives (FP) represent the positive samples incorrectly predicted by the model, and false negatives (FN) represent the negative samples incorrectly predicted by the model; $AP_k$ represents the AP of class k; n represents the number of classes. The P–R curves were also used for visually demonstrating the performance of the models, where P and R were shown on vertical and horizontal axes, respectively. The Intersection over Union (IoU) and the confidence score were both set to 0.5 for test set. Additionally, $R^2$ and root mean square error (RMSE; Equation (9)) was adopted to compare the results predicted by the models and ground truth data from both manual labeling and in-field manual counting:

$$RMSE = \sqrt{\frac{\sum_{i=1}^{N}(x_i - \hat{x}_i)^2}{N}} \tag{9}$$

where i represents one data point (a plant), N represents the number of data points (plants), $x_i$ represents the actual count of grape bunches (in-field or label), and $\hat{x}_i$ represents the estimated count of grape bunches using Swin-T-YOLOv5.

**Table 3.** Major hyper-parameters used in this study for YOLOv3, YOLOv4, YOLOv5, and Swin-transformer-YOLOv5 (Swin-T-YOLOv5).

| Hyper-Parameter | Faster R-CNN | YOLOv3 | YOLOv4 | YOLOv5 | Swin-T-YOLOv5 |
|---|---|---|---|---|---|
| Optimization algorithm | SGD [1] | SGD | SGD | SGD | SGD |
| Initial learn rate | $1 \times 10^{-3}$ | $1 \times 10^{-3}$ | $1 \times 10^{-3}$ | $1 \times 10^{-2}$ | $1 \times 10^{-2}$ |
| Learn rate drop factor | 0.1 | 0.1 | 0.1 | 0.2 | 0.2 |
| Mini-batch size | 256 | 16 | 16 | 32 | 32 |
| Number of epochs | 100 | 100 | 100 | 100 | 100 |
| Intersection over Union (IoU) (train and validation) | 0.3 | 0.213 | 0.213 | 0.2 | 0.2 |
| Weight decay | $5 \times 10^{-4}$ | $5 \times 10^{-4}$ | $5 \times 10^{-4}$ | $5 \times 10^{-4}$ | $5 \times 10^{-4}$ |
| Trainable parameters ($\times 10^6$) | 36.1 | 61.5 | 63.4 | 7.0 | 7.2 |
| Training time | 11 h 18 min | 17 h 35 min | 20 h 48 min | 17 h 50 min | 18 h 4 min |

[1] SGD refers to stochastic gradient descent.

## 3. Results

### 3.1. Swin-Transformer-YOLOv5 Training and Validation

All models were trained and validated with the same dataset for a comprehensive comparison. Table 4 shows the detailed comparison results using the previously defined evaluation metrics. Overall, our proposed Swin-T-YOLOv5 outperformed all other tested models, with an mAP of 97.4%, an F1-score of 0.96, and an inference speed of 13.2 ms per image. The mAP of Swin-T-YOLOv5 was 34.8%, 2.1%, 3.2%, and 2.1% better than that of Faster R-CNN, YOLOv3, YOLOv4, and YOLOv5, respectively. Although its inference time was slightly slower (~12%) than the original YOLOv5, it was still faster than the rest of the models by 0.6–336.8 ms per image. Moreover, P–R curves (Figure 5) showed that Swin-T-YOLOv5 had the best performance among all models as it reached the furthest at the top-right corner (blue curve). Faster R-CNN (in yellow color) performed the worst among the models tested.

**Table 4.** Comparison of training and validation results for Faster R-CNN, YOLOv3, YOLOv4, YOLOv5, and Swin-transformer-YOLOv5 (Swin-T-YOLOv5).

| Network | Precision (%) | Recall (%) | mAP$_{IoU=0.5}$ (%) [1] | F1-Score | Inference Speed per Image (ms) [2] |
|---|---|---|---|---|---|
| Faster R-CNN | 60.1 | 59.1 | 62.6 | 0.59 | 350.0 |
| YOLOv3 | 96.1 | 93.4 | 95.3 | 0.94 | 13.8 |
| YOLOv4 | 75.5 | 92.8 | 94.2 | 0.82 | 14.2 |
| YOLOv5 | 96.6 | 91.4 | 95.3 | 0.94 | 11.8 |
| Swin-T-YOLOv5 | 97.9 | 94.7 | 97.4 | 0.96 | 13.2 |

[1] mAP refers to mean average precision. [2] Inference time may vary depending on the hardware configurations.

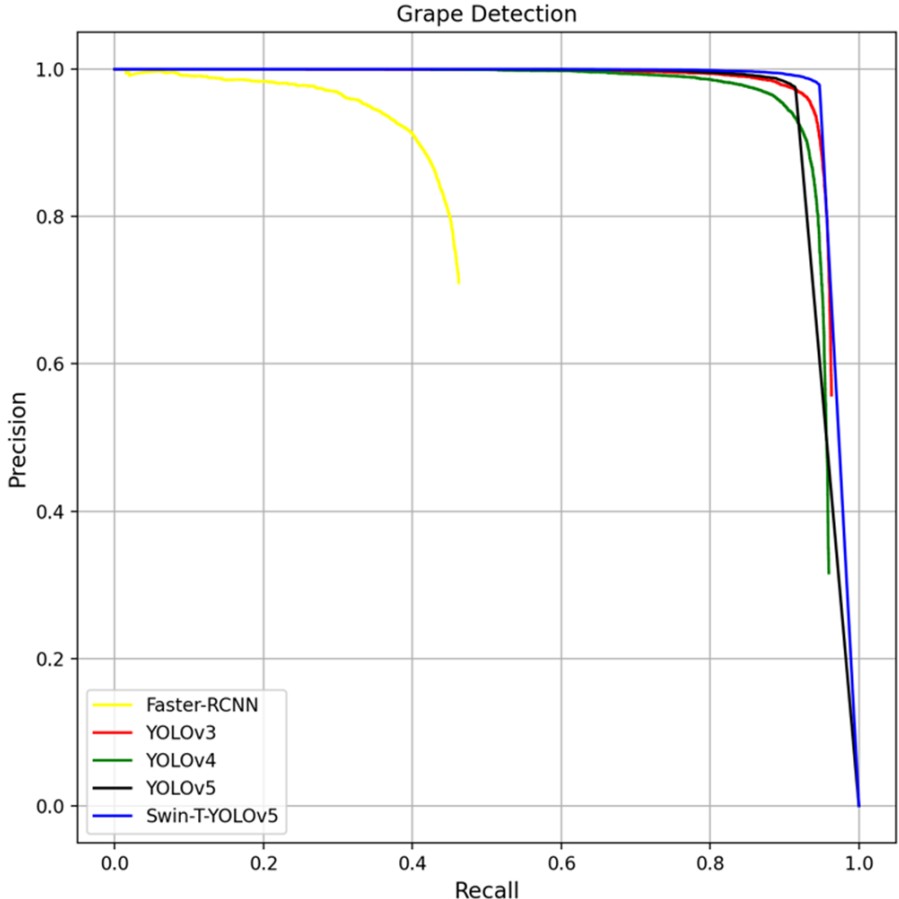

**Figure 5.** Precision–recall (P–R) curves of Faster R-CNN, YOLOv3, YOLOv4, YOLOv5, and Swin-transformer-YOLOv5 (Swin-T-YOLOv5) in detecting grape bunches.

### 3.2. Swin-Transformer-YOLOv5 Model Testing

#### 3.2.1. Testing under Two Weather Conditions

All models were tested under different conditions as listed in Table 1, including two weather conditions (i.e., sunny and cloudy), two berry maturity stages (i.e., immature and mature), and three sunlight directions/intensities (i.e., morning, noon, and afternoon), to verify the superiority of the proposed Swin-T-YOLOv5. Detailed model comparison results were given in Table 5 using the test set under two weather conditions for both grape varieties of Chardonnay and Merlot. Compared to Faster R-CNN, YOLOv3, YOLOv4, and YOLOv5, Swin-T-YOLOv5 achieved the best performance under both conditions in terms of mAP (95.4–97.2%) and F1-score (0.86–0.89). Swin-T-YOLOv5 performed slightly better under cloudy sky conditions with higher mAP (+1.8%) and F1-score (+0.03) compared to

the same measures under sunny conditions. While the inference speed of Swin-T-YOLOv5 (13.2 ms per image) was not the best among all, it was 1.4 ms slower than YOLOv5 only.

**Table 5.** Model comparison using the test set under two weather conditions.

| Model | Weather Condition | $mAP_{IoU=0.5}$ (%) [1] | F1-Score | Inference Speed per Image (ms) [2] |
|---|---|---|---|---|
| Faster R-CNN | | 59.23 | 0.63 | 350 |
| YOLOv3 | | 84.47 | 0.64 | 13.8 |
| YOLOv4 | Sunny | 90.43 | 0.68 | 14.2 |
| YOLOv5 | | 92.16 | 0.82 | 11.8 |
| Swin-T-YOLOv5 | | 95.36 | 0.86 | 13.2 |
| Faster R-CNN | | 53.54 | 0.67 | 350 |
| YOLOv3 | | 78.93 | 0.72 | 13.8 |
| YOLOv4 | Cloudy | 83.45 | 0.76 | 14.2 |
| YOLOv5 | | 93.64 | 0.83 | 11.8 |
| Swin-T-YOLOv5 | | 97.19 | 0.89 | 13.2 |

[1] mAP refers to mean average precision. [2] Inference time may vary depending on the hardware configurations.

As it was proven that Swin-T-YOLOv5 outperformed all other tested models under both sunny and cloudy sky conditions, we further compared it against the ground truth data from manual labeling and in-field manual counting (Figure 6) for Chardonnay and Merlot, respectively. Results showed that Swin-T-YOLOv5 performed well with Chardonnay variety under both weather conditions with 0.70–0.82 of $R^2$ and 2.9–5.1 RMSE when the predicted results were compared against both in-field and label counts (Figure 6a–c). It also worked well with Merlot under cloudy conditions (Figure 6d), however, $R^2$ dropped to 0.28–0.36 and RMSE increased to ~7.0 on Merlot under sunny condition (Figure 6b), indicating greater detection errors. Demonstrations of detection results under two weather conditions were provided in Figures A1 and A2 in Appendix A.

3.2.2. Testing at Two Maturity Stages

In addition to two different weather conditions, we compared the performances of Swin-T-YOLOv5 with all other studied models at two berry maturity stages, including immature and mature berries for Chardonnay (i.e., white color of berry skin throughout the growing season) and Merlot (i.e., white or white–red mix when immature; red color when mature) (Figure 1). Detailed comparison results were given in Table 6 that, as expected, Swin-T-YOLOv5 outperformed all other tested models at both berry maturity stages with 90.3–95.9% of mAP and 0.82–0.87 of F1-score. Clearly, all detectors achieved better detection results at the mature stage, including Swin-T-YOLOv5 (5.6% higher in mAP and 0.05 higher in F1-score), when the berries were larger (i.e., less occlusions) and with more distinct color than their background, such as leaves. Compared to the second-best model, YOLOv5 (mAPs of 89.8–91.6%), the performance of Swin-T-YOLOv5 was improved more at the berry mature stage (+4.3%) than the immature stage (+0.5%), indicating that the improvements of the model were more effective to those ready-to-harvest grape bunches.

Figure 7 compared the specific predicted number of grape bunches using Swin-T-YOLOv5 against the ground truth data of both manual labeling and in-field manual counting on both Chardonnay and Merlot. As observed in Table 6, $R^2$ was higher (0.57–0.89) and RMSE was smaller (2.5–3.9) for those mature berries (Figure 7c,d). Swin-T-YOLOv5 did a poor job on Merlot when the berries were immature (i.e., white or white–red mixed berries) with 0.08–0.16 of $R^2$ and 8.6–9.0 RMSE (Figure 7b). Demonstrations of detection results at two berry maturity stages were provided in Figures A3 and A4 in Appendix A.

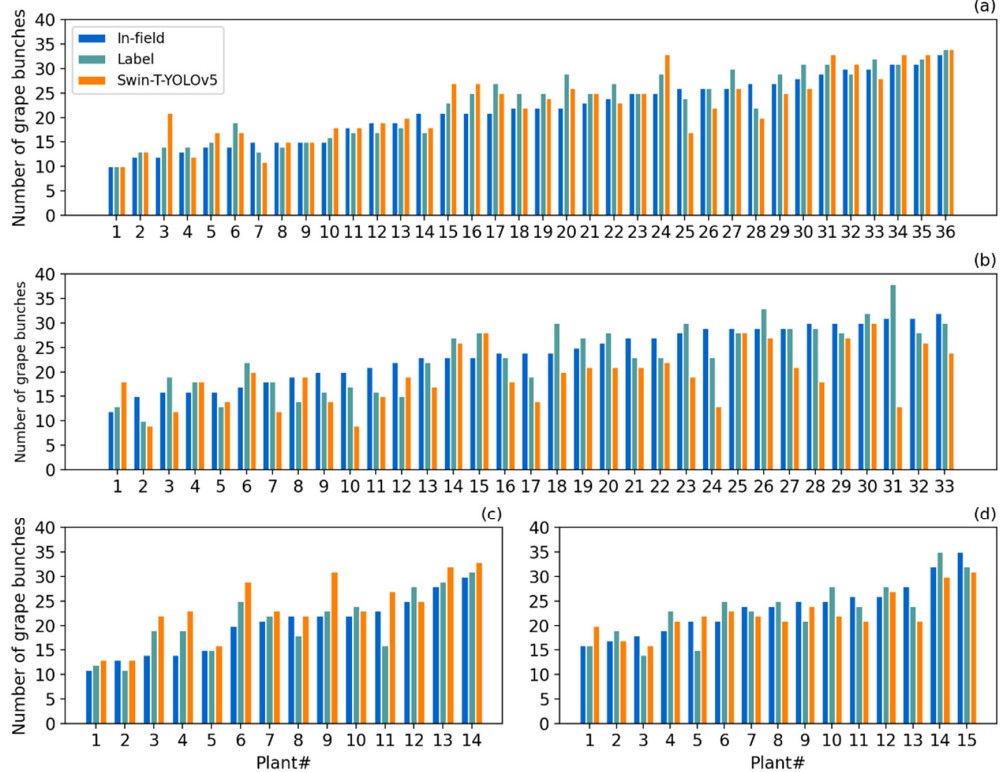

| Statistics | (a) Chardonnay/Sunny | | (b) Merlot/Sunny | | (c) Chardonnay/Cloudy | | (d) Merlot/Cloudy | |
|---|---|---|---|---|---|---|---|---|
| | $R^2$ | RMSE | $R^2$ | RMSE | $R^2$ | RMSE | $R^2$ | RMSE |
| In-field vs. Swin-T-YOLOv5 | 0.70 | 3.71 | 0.28 | 7.04 | 0.70 | 5.05 | 0.69 | 3.13 |
| Label vs. Swin-T-YOLOv5 | 0.82 | 2.93 | 0.36 | 6.97 | 0.72 | 4.40 | 0.66 | 3.54 |

**Figure 6.** The number of grape bunches comparison between in-field manual counting (gTruth), manual label, and detection using Swin-transformer-YOLOv5 (Swin-T-YOLOv5) with (**a**) Chardonnay (sunny); (**b**) Merlot (sunny); (**c**) Chardonnay (cloudy); and (**d**) Merlot (cloudy). RMSE refers to root mean square error. Plant# refers to the number of plants.

**Table 6.** Model comparison using the test set at two berry maturity stages.

| Model | Berry Maturity | mAP$_{IoU=0.5}$ (%) [1] | F1-Score | Inference Speed per Image (ms) [2] |
|---|---|---|---|---|
| Faster R-CNN | | 50.12 | 0.52 | 350 |
| YOLOv3 | | 82.84 | 0.60 | 13.8 |
| YOLOv4 | Immature | 87.24 | 0.65 | 14.2 |
| YOLOv5 | | 89.78 | 0.80 | 11.8 |
| Swin-T-YOLOv5 | | 90.31 | 0.82 | 13.2 |
| Faster R-CNN | | 52.35 | 0.59 | 350 |
| YOLOv3 | | 85.43 | 0.76 | 13.8 |
| YOLOv4 | Mature | 89.40 | 0.77 | 14.2 |
| YOLOv5 | | 91.58 | 0.81 | 11.8 |
| Swin-T-YOLOv5 | | 95.86 | 0.87 | 13.2 |

[1] mAP refers to mean average precision. [2] Inference time may vary depending on the hardware configurations.

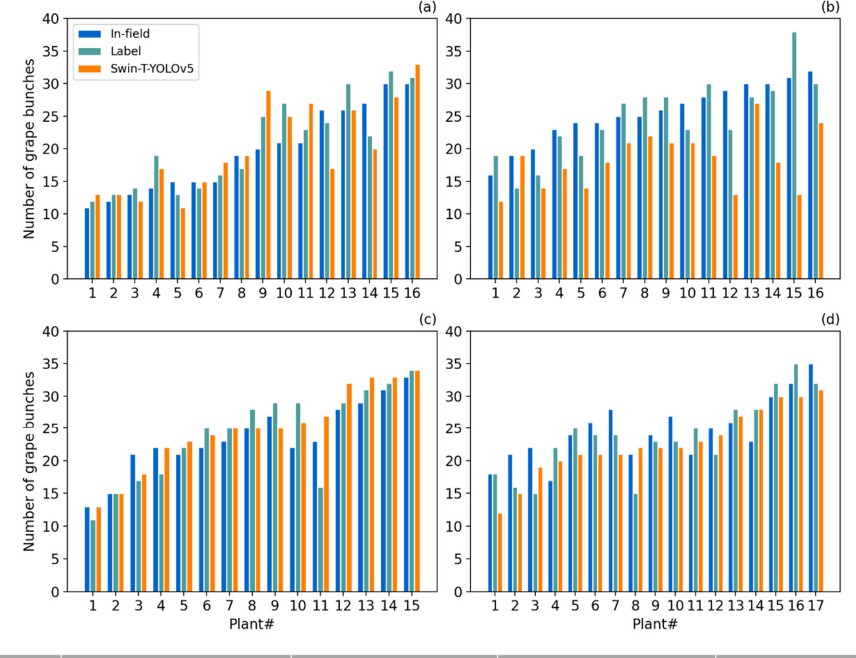

| **Statistics** | **(a) Chardonnay/Immature** | | **(b) Merlot/Immature** | | **(c) Chardonnay/Mature** | | **(d) Merlot/Mature** | |
|---|---|---|---|---|---|---|---|---|
| | $R^2$ | RMSE | $R^2$ | RMSE | $R^2$ | RMSE | $R^2$ | RMSE |
| In-field vs. Swin-T-YOLOv5 | 0.59 | 4.44 | 0.16 | 8.61 | 0.89 | 2.50 | 0.57 | 3.86 |
| Label vs. Swin-T-YOLOv5 | 0.81 | 3.03 | 0.08 | 8.95 | 0.74 | 3.57 | 0.70 | 3.30 |

**Figure 7.** The number of grape bunches comparison between in-field manual counting (gTruth), manual label, and detection using Swin-transformer-YOLOv5 (Swin-T-YOLOv5) for (**a**,**b**) immature and (**c**,**d**) mature berries with Chardonnay (left) and Merlot (right). RMSE refers to root mean square error. Plant# refers to the number of plants.

### 3.2.3. Testing under Three Sunlight Directions and Intensities

Finally, all models were tested under three different sunlight directions and intensities, including in the morning (8am–9am), noon (11am–12pm), and afternoon (4pm–5pm) (Table 1). The light intensity was highest at noon and lowest in the morning. Specific comparison results were given in Table 7. Among all models tested in this research, Swin-T-YOLOv5 performed the best under any sunlight condition, with optimal mAPs of 92.0–94.5% and F1-scores of 0.83–0.86. It was also obvious that the detection results were better at noon than in the morning or afternoon with 2.5–2.6% higher mAP and 0.01–0.03 higher F1-score. Additionally, YOLOv5 still performed the second best except for during noon, where Swin-T-YOLOv5 and YOLOv4 achieved 6.1% and 1.7% better mAP than it.

**Table 7.** Model comparison using the test set under three sunlight directions and intensities.

| Model | Sunlight Condition | $mAP_{IoU=0.5}$ (%) [1] | F1-Score | Inference Speed per Image (ms) [2] |
|---|---|---|---|---|
| Faster R-CNN | | 55.35 | 0.56 | 350 |
| YOLOv3 | | 74.57 | 0.65 | 13.8 |
| YOLOv4 | Morning | 78.15 | 0.67 | 14.2 |
| YOLOv5 | | 89.57 | 0.79 | 11.8 |
| Swin-T-YOLOv5 | | 92.04 | 0.83 | 13.2 |

**Table 7.** *Cont.*

| Model | Sunlight Condition | $mAP_{IoU=0.5}$ (%) [1] | F1-Score | Inference Speed per Image (ms) [2] |
|---|---|---|---|---|
| Faster R-CNN | | 60.73 | 0.59 | 350 |
| YOLOv3 | | 86.31 | 0.67 | 13.8 |
| YOLOv4 | Noon | 90.16 | 0.70 | 14.2 |
| YOLOv5 | | 88.45 | 0.79 | 11.8 |
| Swin-T-YOLOv5 | | 94.53 | 0.86 | 13.2 |
| Faster R-CNN | | 56.79 | 0.52 | 350 |
| YOLOv3 | | 76.78 | 0.70 | 13.8 |
| YOLOv4 | Afternoon | 81.12 | 0.73 | 14.2 |
| YOLOv5 | | 87.46 | 0.80 | 11.8 |
| Swin-T-YOLOv5 | | 91.96 | 0.85 | 13.2 |

[1] mAP refers to mean average precision. [2] Inference time may vary depending on the hardware configurations.

Further observations on the number of grape bunches detected by Swin-T-YOLOv5 comparing against ground truth data, from manual labeling and in-field manual counting, were provided in Figure 8. For the Chardonnay variety, the agreement between the predictions and ground truth was relatively better (0.55–0.91 of $R^2$ and 2.4–4.7 of RMSE; Figure 8a,c,e) than the Merlot variety (0.13–0.70 of $R^2$ and 5.1–7.1 of RMSE; Figure 8b,d,f) under any sunlight conditions. The results for Merlot were the best at noon with 0.47–0.70 of $R^2$ (Figure 8d), while the results were the worst in the afternoon with only 0.13–0.29 of $R^2$ when the imaging side was against the direction of the sunlight (Figure 8f). Visual comparisons of model performances under different sunlight conditions can be found in Figures A5 and A6 in Appendix A.

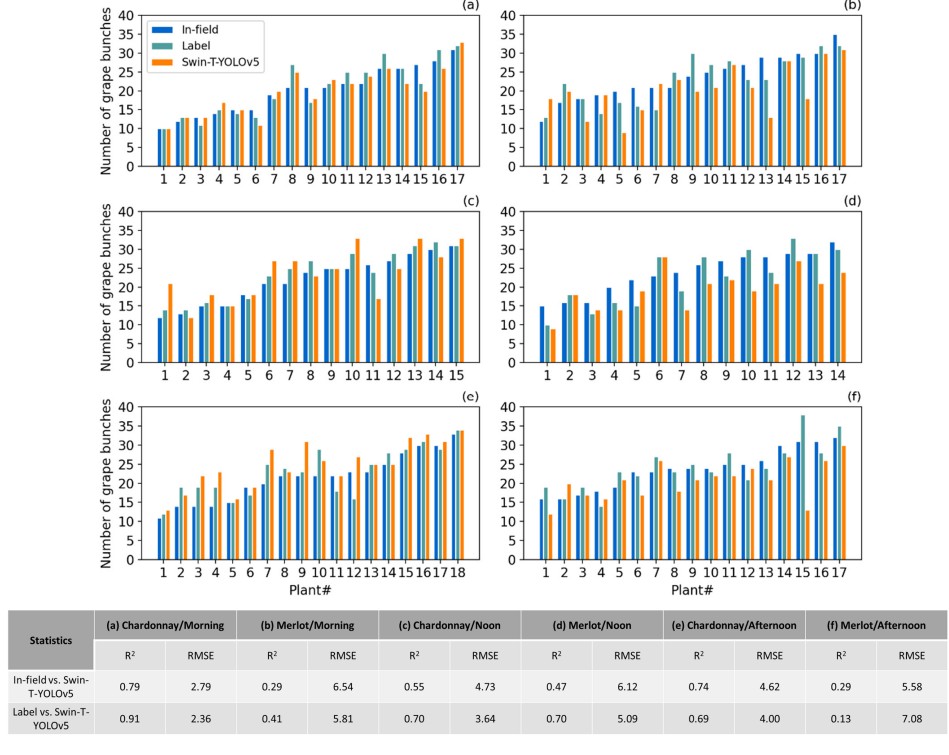

**Figure 8.** The number of grape bunches comparison between in-field manual counting (gTruth), manual label, and detection using Swin-transformer-YOLOv5 (Swin-T-YOLOv5) during (**a**,**b**) morning; (**c**,**d**) noon; (**e**,**f**) afternoon with Chardonnay (left) and Merlot (right). RMSE refers to root mean square error. Plant# refers to the number of plants.

## 4. Discussion

Compared to other fruits, such as apple, citrus, pear, and kiwifruit, grape bunches in the vineyards have more complex structural shapes and silhouette characteristics to be accurately detected using machine vision systems. Accurate and fast identification of overlapped grape bunches in dense foliage canopies under natural lighting environments remains to be a key challenge in vineyards. Therefore, this research proposed the combination of architectures from a conventional CNN YOLOv5 model and a Swin-transformer model that can inherit the advantages of both models to preserve global and local features when detecting grape bunches (Figure 4). The newly integrated detector (i.e., Swin-T-YOLOv5) worked as expected in overcoming the drawbacks of CNNs in capturing the global features (Figure 5).

Our proposed Swin-T-YOLOv5 was tested on two different wine grape varieties, two different weather/sky conditions, two different berry maturity stages, and three different sunlight directions/intensities for its detection performance (Table 1). A comprehensive evaluation was made by comparing Swin-T-YOLOv5 against various commonly used detectors (Table 4). Results verified that our proposed Swin-T-YOLOv5 outperformed all other tested models under any listed environmental conditions with achieved 90.3–97.2% mAPs. The best and worst results were obtained under cloudy weather and berry immature conditions, respectively, with a 6.9% in difference.

Specifically, Swin-T-YOLOv5 performed the best under cloudy weather conditions with the highest mAP of 97.2%, which was 1.8% higher than in sunny weather conditions (Table 5), although the difference was inconsiderable. While testing the models at different berry maturity stages, Swin-T-YOLOv5 performed much better when the berries were mature with 95.9% of mAP than immature berries (with 5.6% lower mAP; Table 6). It has been verified that crop early thinning can provide more berry quality benefits than late thinning in vineyards [51]. Therefore, early (immature stage) grape bunch detection is more meaningful than late (mature stage) detection in our study. However, early detection is challenging because the berries tended to be smaller in size and lighter in color during the early growth stage and thus more difficult to be detected. Moreover, Swin-T-YOLOv5 achieved better mAP at noon (94.5%) than the other two timings in the day (Table 7), while the afternoon sunlight condition more negatively affected the model with a lower mAP of 92.0% than in the morning. Apparently, the effectiveness of the berry maturities and light directions can be the major reasons for impacting the performances of the models, while weather conditions almost did not change the detection results. The improvements from the original YOLOv5 to the proposed Swin-T-YOLOv5 varied based on the conditions (0.5–6.1%), however, the maximum increment happened when comparing them at noon (Table 7). Overall, it was confirmed that the Swin-T-YOLOv5 achieved the best results in terms of mAP (up to 97.2%) and F1-score (up to 0.9) among all compared models in this research for wine grape bunch detections in vineyards. Its inference speed was the second best (13.2 ms per image) only after YOLOv5's (11.8 ms per image) under any test conditions.

To further assess the model performance, we compared the predicted number of grape bunches by Swin-T-YOLOv5 with both manual labeling and in-field manual counting. The $R^2$ and RMSE between Swin-T-YOLOv5 and in-field counting had the similar trends of the ones between Swin-T-YOLOv5 and manual labeling in general, potentially because some of those heavily occluded grape bunches were not taken into consideration for labeling during the annotation process. However, the values changed vastly for the two different grape varieties under various conditions. It was clear that Swin-T-YOLOv5 did not perform well on the Merlot variety when the weather condition was sunny. While for Chardonnay, the model performed well under either condition (Figure 6). Similarly, the performance of Swin-T-YOLOv5 was poor on the Merlot variety when the berries were immature. For Chardonnay, Swin-T-YOLOv5 achieved better results under either condition (Figure 7). In addition, Swin-T-YOLOv5 underperformed on the Merlot variety when the sunlight condition was in the afternoon. Comparatively, it worked better on Merlot at noon. For

Chardonnay, the predictions were more accurate (Figure 8). This was possibly because the Merlot variety had a more complex combination of grape bunches when the berries were immature with either white or white–red mixed color (Figure 1e), which may cause more detection errors under more challenging test conditions, such as when imaging against the direction of the light. In general, detecting grape bunches of the Merlot variety was more challenging than the Chardonnay variety under any test conditions in this research.

We found that our proposed Swin-T-YOLOv5 could enhance the accuracy of grape bunch detection when the grape bunches were slightly/moderately occluded or clustered, attributed to the Swin-transformer module that was added to the generic YOLOv5. For example, slight/moderate canopy occlusions and overlap of grape clusters can cause detection errors in terms of detected number of bounding boxes, i.e., underestimations in Figure 9a,b, overestimation in Figure 9c, and bounding box misplace in Figure 9d, comparing the results from YOLOv5 and Swin-T-YOLOv5. Swin-transformer module assisted in detecting the objects in line with common sense under such conditions. Clearly, introducing the self-attention mechanism in the backbone network should be a right direction.

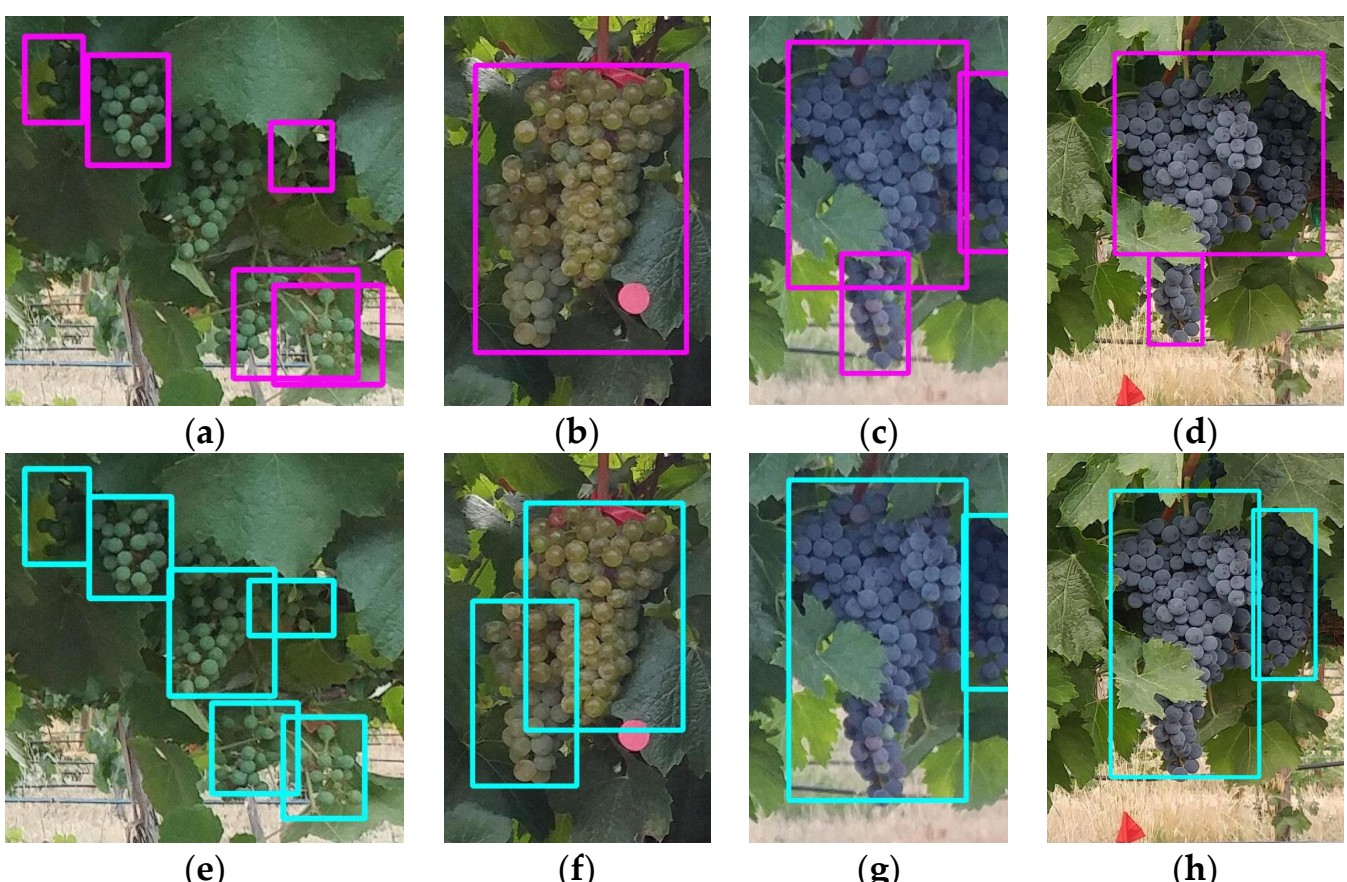

**Figure 9.** Detection results on Chardonnay (white variety; two columns on the left) and Merlot (red variety; two columns on the right) using (**a–d**) generic YOLOv5 (in magenta color) and (**e–h**) Swin-transformer-YOLOv5 (Swin-T-YOLOv5; in cyan color) in zoomed-in views.

Although Swin-T-YOLOv5 outperformed all other tested models in detecting grape bunches under various external or internal variations, detection failures (i.e., TNs and FPs) happened more frequently in several scenarios as illustrated in Figure 10. For example, severe occlusion (mainly by leaves) caused detection failure, which was the major reason for having TNs and FPs in this research as marked out using the red bounding boxes, particularly when the visible part of grape bunches were small (Figure 10e,f) or having

the similar color compared to the background (Figure 10a–c). In addition, clustered grape bunches can cause detection failures, where two grape bunches were detected as one (Figure 10d).

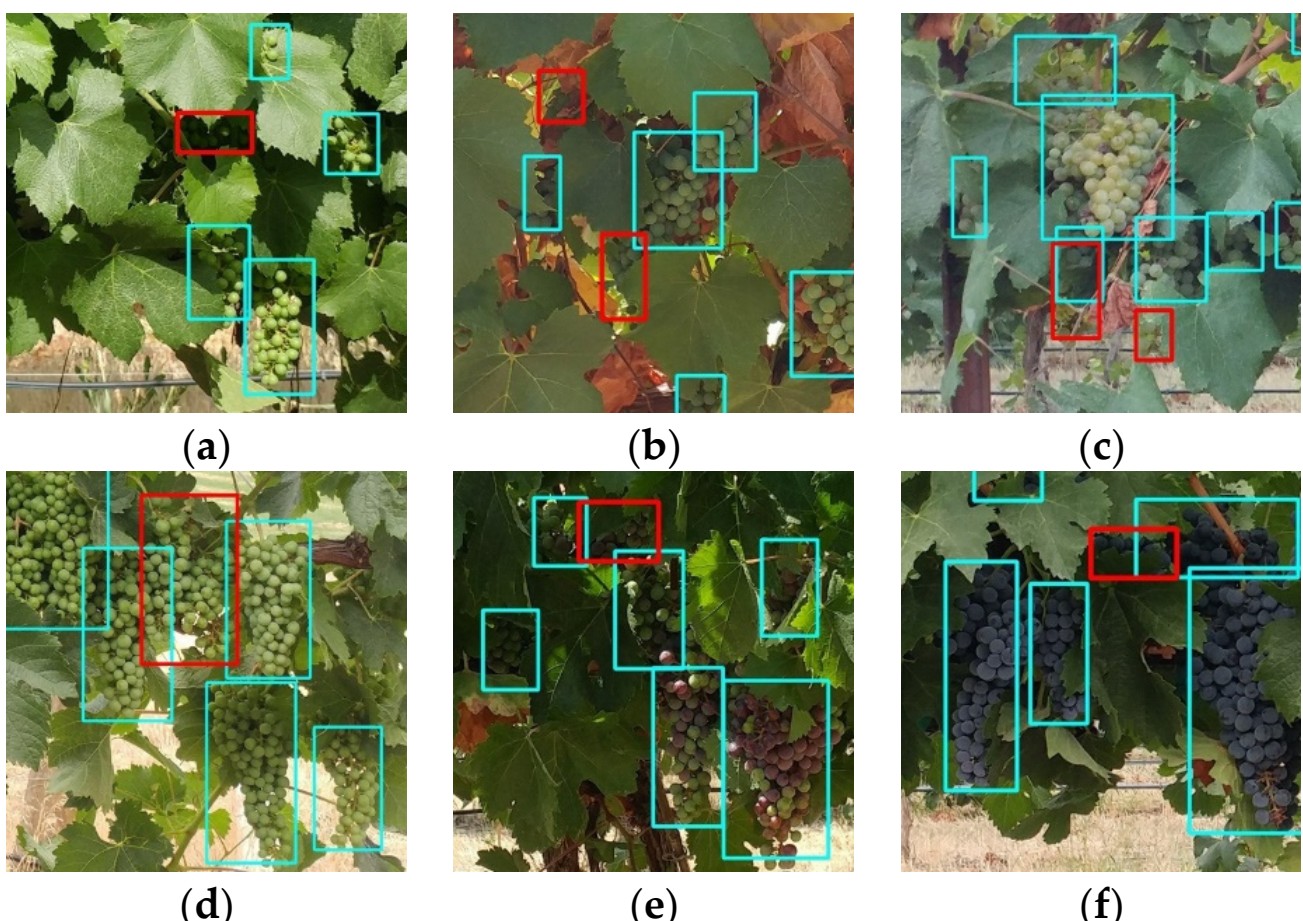

**Figure 10.** Illustrations of failures (i.e., true negatives (TN) or false positives (FP) highlighted in red bounding boxes; true positives (TP) in cyan bounding boxes) in grape bunch detection using Swin-transformer-YOLOv5 (Swin-T-YOLOv5) during early (left column), mid- (middle column), and harvest-stage (right column) for (**a**–**c**) "Chardonnay (white variety)" and (**d**–**f**) "Merlot (red variety)" in zoomed-in views.

## 5. Conclusions

This research proposed an optimal and real-time wine grape bunch detection model in natural vineyards by architecturally integrating YOLOv5 and Swin-transformer detectors, called Swin-T-YOLOv5. The research was carried out on two different grape varieties, Chardonnay (white color of berry skin when mature) and Merlot (red color of berry skin when mature), throughout the growing season from 4 July 2019 to 30 September 2019 under various testing conditions, including two different weather/sky conditions (i.e., sunny and cloudy), two different berry maturity stages (i.e., immature and mature), and three different sunlight directions/intensities (i.e., morning, noon, and afternoon). Further assessment was made by comparing the proposed Swin-T-YOLOv5 with other commonly used detectors, including Faster R-CNN, YOLOv3, YOLOv4, and YOLOv5. Based on the obtained results, the following conclusions can be drawn:

1. Validation results verified the advancement of proposed Swin-T-YOLOv5 with the best precision of 98%, recall of 95%, mAP of 97%, and F1-score of 0.96;
2. Swin-T-YOLOv5 outperformed all other studied models under all test conditions in this research:
3. Two weather conditions: During sunny days, Swin-T-YOLOv5 achieved 95% of mAP and 0.86 of F1-score, which were up to 11% and 0.22 higher than others. During cloudy days, it achieved 97% of mAP and 0.89 of F1-score, which were up to 18% and 0.17 higher;
4. Two berry maturity stages: With immature berries, Swin-T-YOLOv5 achieved 90% of mAP and 0.82 of F1-score, which were up to 7% and 0.22 higher than others. With mature berries, it achieved 96% of mAP and 0.87 of F1-score, which were up to 10% and 0.11 higher;
5. Three sunlight directions/intensities: In the morning, Swin-T-YOLOv5 achieved 92% of mAP and 0.83 of F1-score, which were up to 17% and 0.18 higher than others. At noon, it achieved 95% of mAP and 0.86 of F1-score, which were up to 8% and 0.19 higher; In the afternoon, it achieved 92% of mAP and 0.85 of F1-score, which were up to 15% and 0.15 higher;
6. Swin-T-YOLOv5 performed differently on Chardonnay and Merlot varieties when comparing the predictions against the ground truth data (i.e., manual labeling and in-field manual counting). For the Chardonnay variety, Swin-T-YOLOv5 provided desired predictions under almost all test conditions, with up to 0.91 of $R^2$ and 2.4 of RMSE. For the Merlot variety, Swin-T-YOLOv5 performed better under several test conditions (e.g., 0.70 of $R^2$ and 3.3 of RMSE for mature berries), while underperformed when detecting immature berries (0.08 of $R^2$ and 9.0 of RMSE).

A novel grape bunch detector, Swin-T-YOLOv5, proposed in this study has been verified for its superiority in terms of detection accuracy and inference speed. It is expected that this integrated detection model can be deployed and implemented on portable devices, such as smartphones, to assist wine grape growers for real-time precision vineyard canopy management. Our next steps include (1) designing and developing a front-end user interface (e.g., mobile application) and a back-end program to run the trained Swin-T-YOLOv5 model and (2) establishing a digital dataset repository on GitHub to further enlarge the image dataset specifically for grape canopies.

**Author Contributions:** Conceptualization, S.L. and X.Z.; methodology, S.L., X.L. and X.Z.; software, X.L. and Z.H.; validation, X.L. and X.Z.; formal analysis, X.L., Z.H. and X.Z.; investigation, X.Z. and W.L.; resources, X.Z. and M.K.; data curation, X.L. and Z.H.; writing—original draft preparation, S.L., X.L. and X.Z.; writing—review and editing, Z.H., X.Z., W.L. and M.K.; visualization, X.L. and Z.H.; supervision, X.Z.; project administration, X.Z. All authors have read and agreed to the published version of the manuscript.

**Funding:** This research received no external funding.

**Data Availability Statement:** Publicly available datasets were analyzed in this study. Upon acceptance of this manuscript by the journal, this data can be found here: https://github.com/LiuXiaoYu2030/The-Grape-Dataset (accessed on 19 October 2022).

**Conflicts of Interest:** The authors declare no conflict of interest.

**Appendix A**

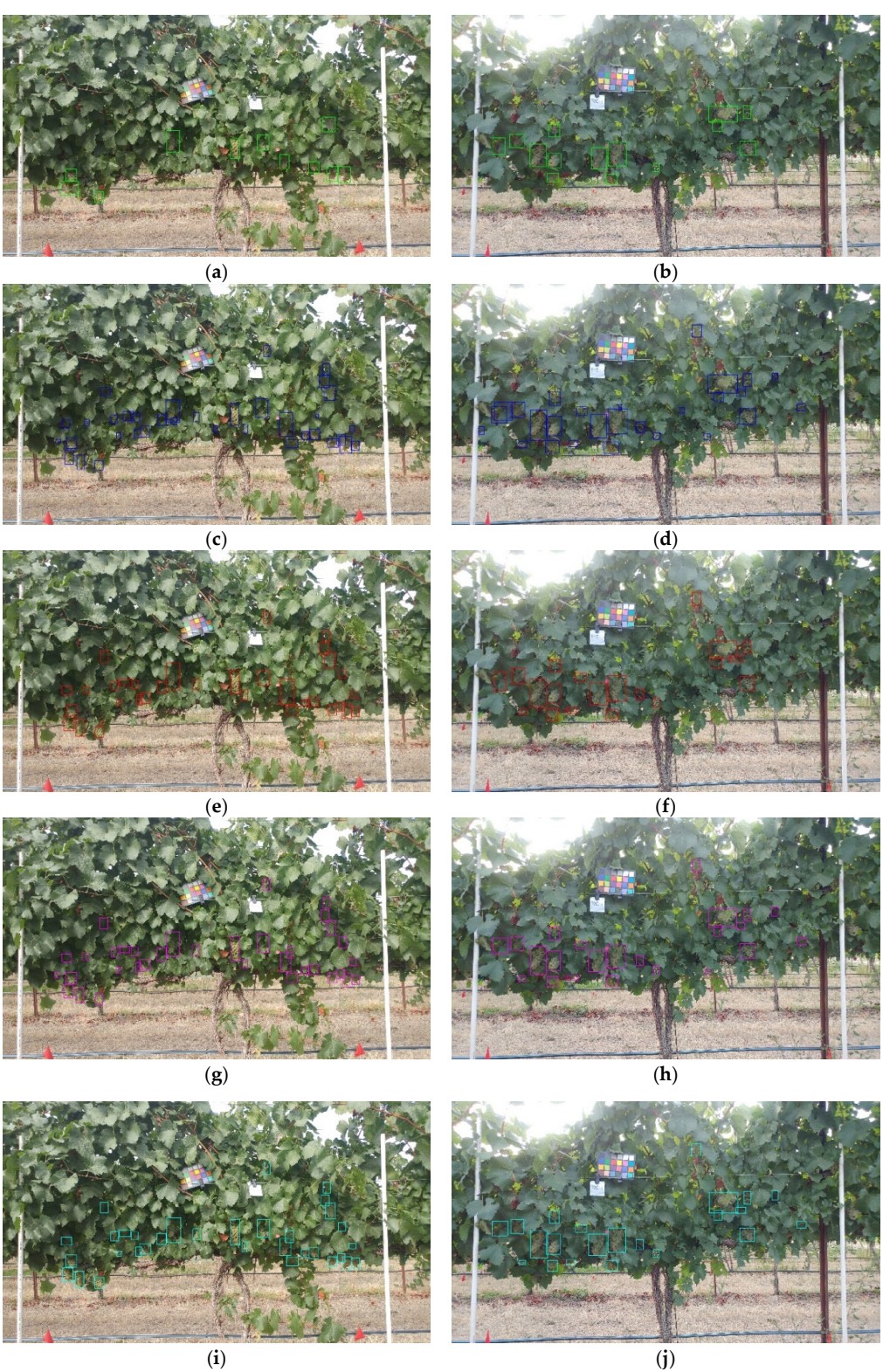

**Figure A1.** Demonstrations of detection results on the test set of Chardonnay (white variety) using (**a**,**b**) Faster R-CNN (bounding boxes in green color); (**c**,**d**) YOLOv3 (in blue color); (**e**,**f**) YOLOv4 (in red color); (**g**,**h**) YOLOv5 (in magenta color); and (**i**,**j**) Swin-transformer-YOLOv5 (in cyan color) under sunny (left) and cloudy (right) weathers.

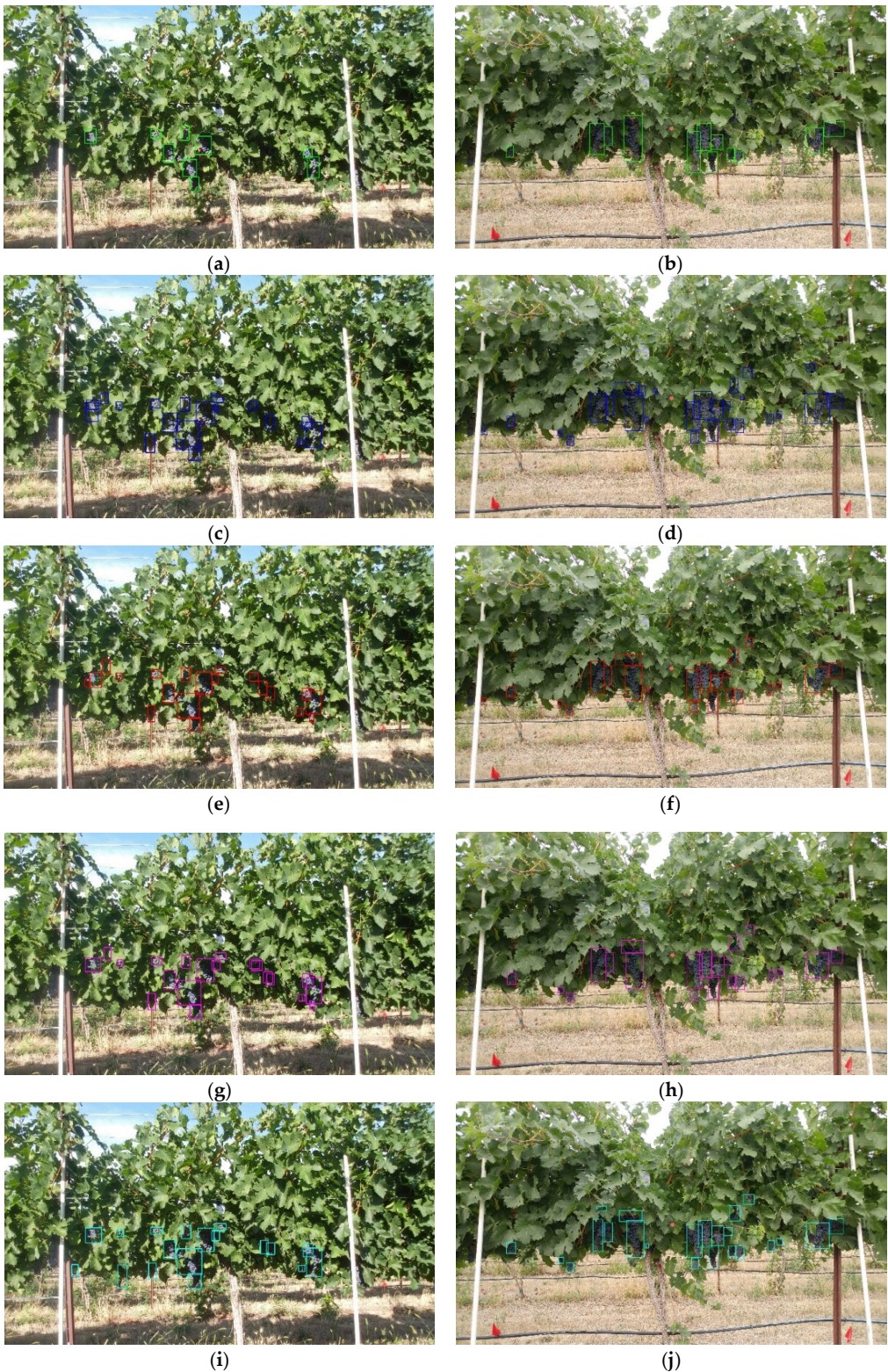

**Figure A2.** Demonstrations of detection results on the test set of Merlot (red variety) using (**a**,**b**) Faster R-CNN (bounding boxes in green color); (**c**,**d**) YOLOv3 (in blue color); (**e**,**f**) YOLOv4 in red color; (**g**,**h**) YOLOv5 (in magenta color); and (**i**,**j**) Swin-transformer-YOLOv5 (in cyan color) under sunny (left) and cloudy (right) weathers.

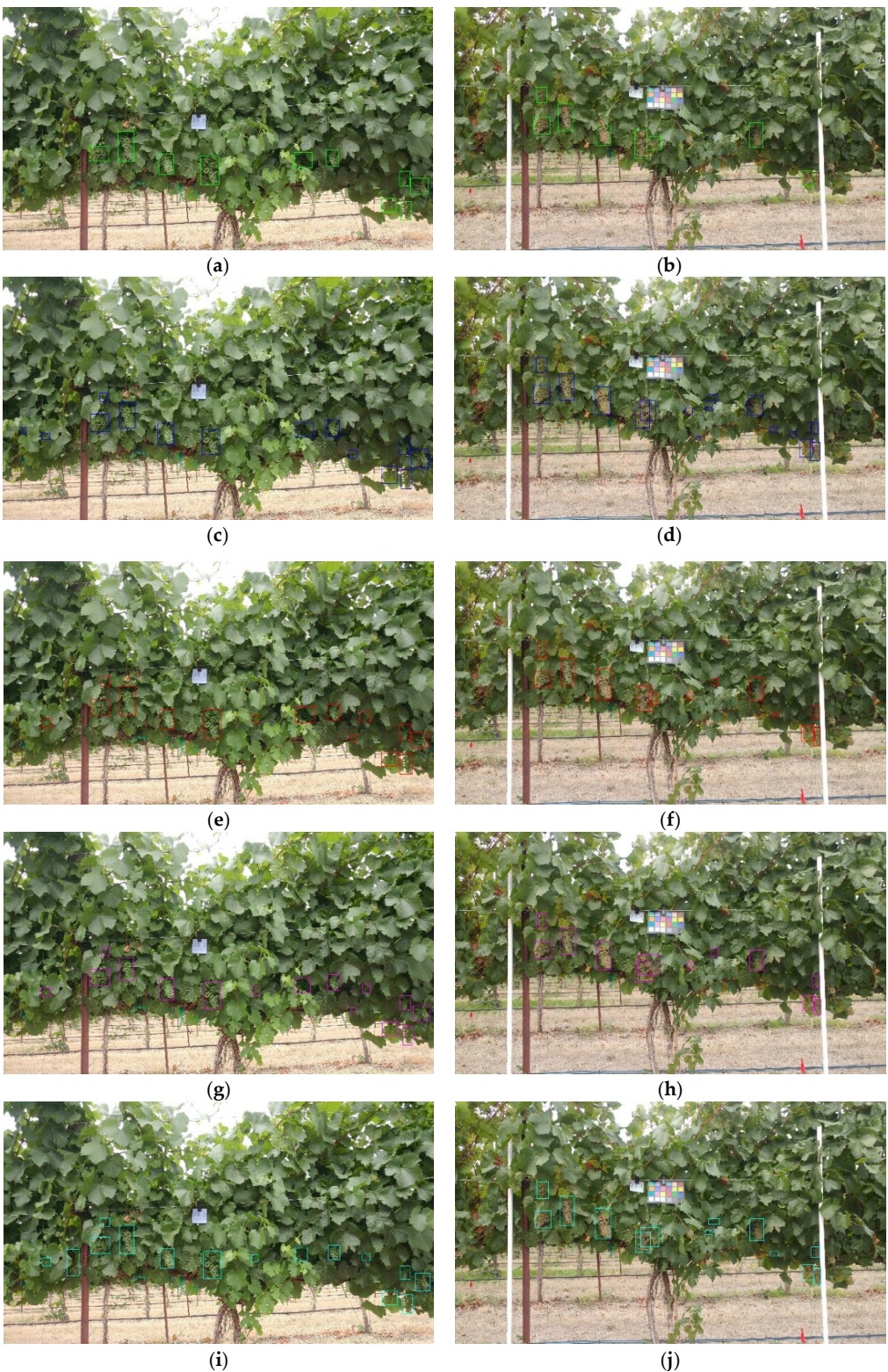

**Figure A3.** Demonstrations of detection results on the test set of Chardonnay (white variety) using (**a**,**b**) Faster R-CNN (bounding boxes in green color); (**c**,**d**) YOLOv3 (in blue color); (**e**,**f**) YOLOv4 (in red color); (**g**,**h**) YOLOv5 (in magenta color); and (**i**,**j**) Swin-transformer-YOLOv5 (in cyan color) at immature (left) and mature (right) stages.

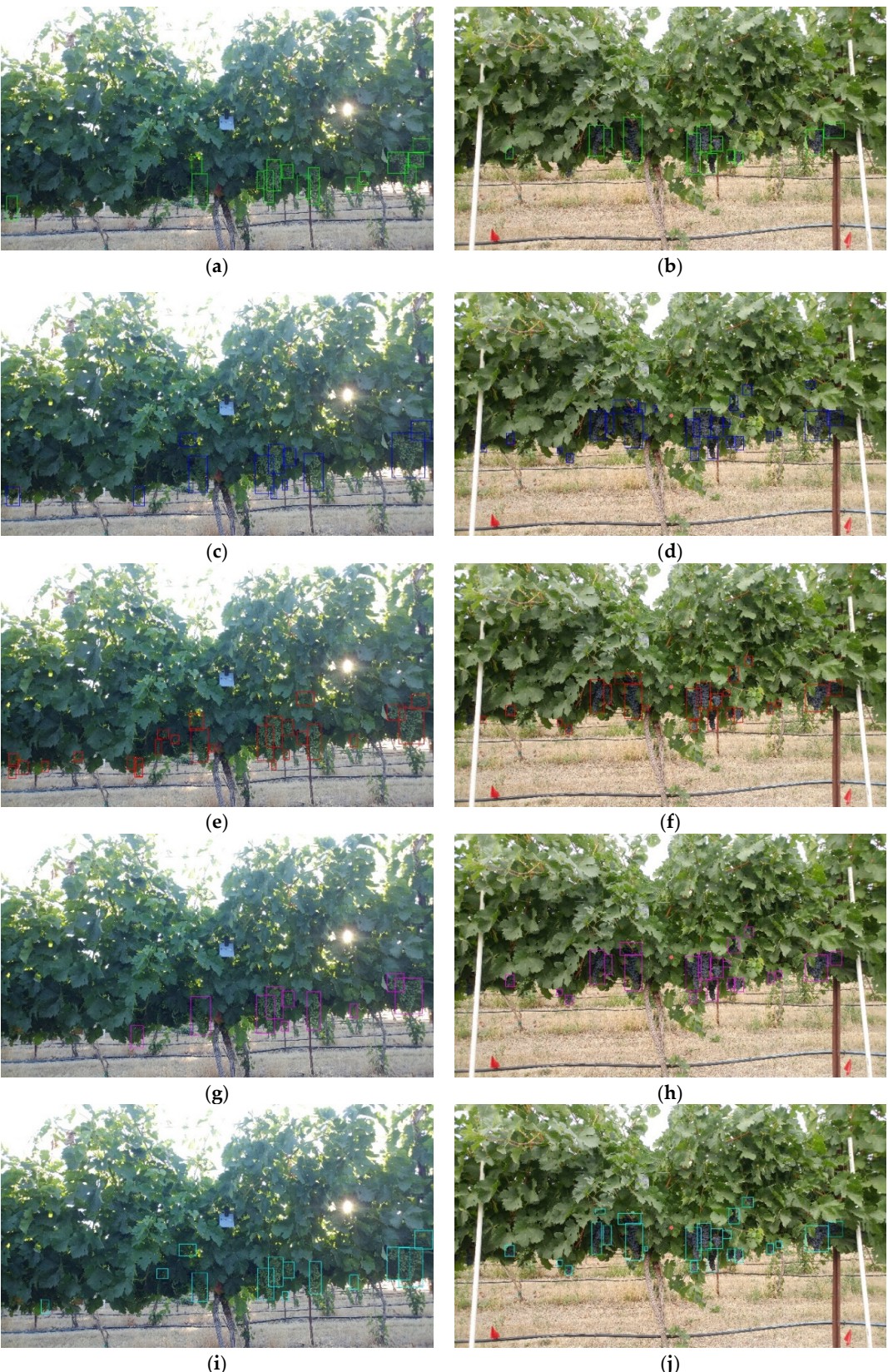

**Figure A4.** Demonstrations of detection results on the test set of Merlot (red variety) using (**a**,**b**) Faster R-CNN (bounding boxes in green color); (**c**,**d**) YOLOv3 (in blue color); (**e**,**f**) YOLOv4 (in red color); (**g**,**h**) YOLOv5 (in magenta color); and (**i**,**j**) Swin-transformer-YOLOv5 (in cyan color) at immature (left) and mature (right) stages.

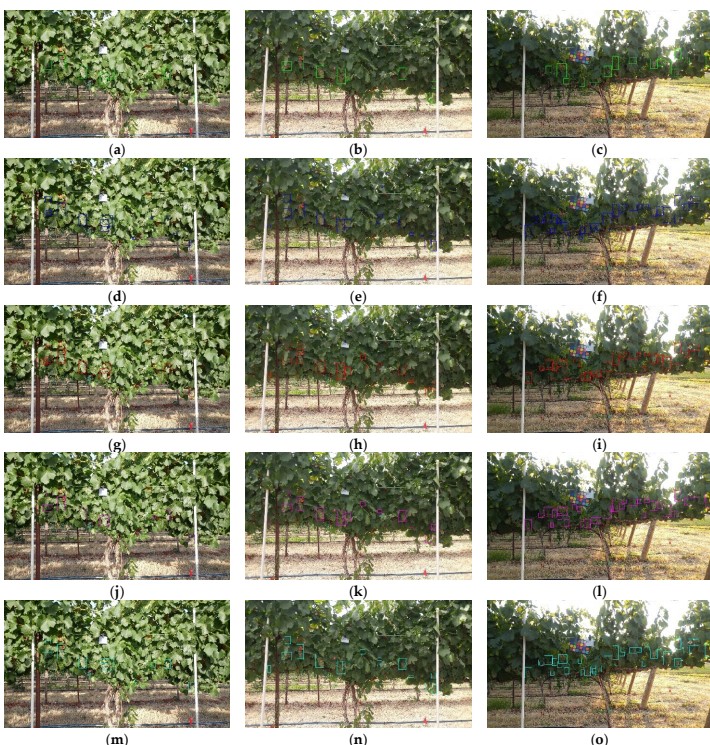

**Figure A5.** Demonstrations of detection results on the test set of Chardonnay (white variety) using (**a**–**c**) Faster R-CNN (bounding boxes in green color); (**d**–**f**) YOLOv3 (in blue color); (**g**–**i**) YOLOv4 (in red color); (**j**–**l**) YOLOv5 (in magenta color); and (**m**–**o**) Swin-transformer-YOLOv5 (in cyan color) under morning (left), noon (middle), and afternoon (right) sunlight directions/intensities.

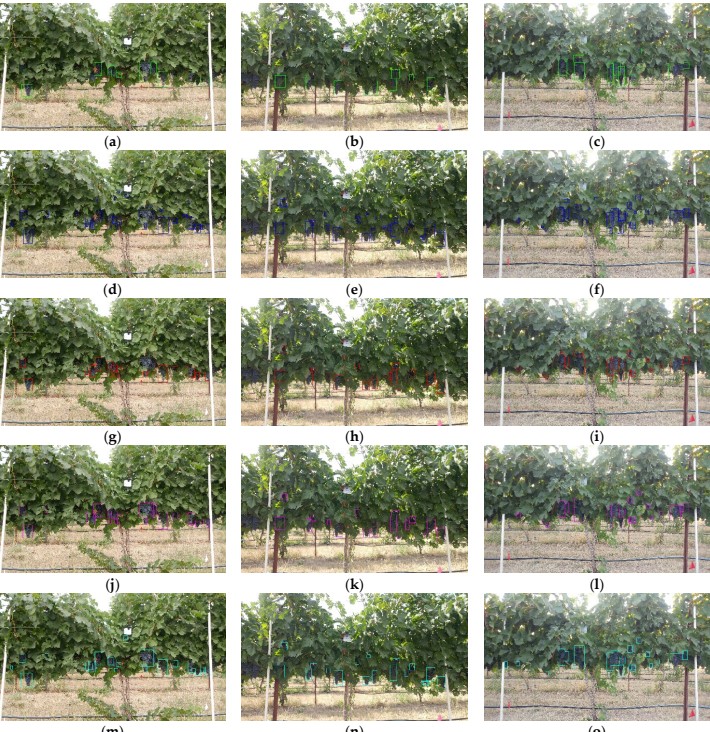

**Figure A6.** Demonstrations of detection results on the test set of Merlot (red variety) using (**a**–**c**) Faster R-CNN (bounding boxes in green color); (**d**–**f**) YOLOv3 (in blue color); (**g**–**i**) YOLOv4 (in red color); (**j**–**l**) YOLOv5 (in magenta color); and (**m**–**o**) Swin-transformer-YOLOv5 (in cyan color) under morning (left), noon (middle), and afternoon (right) sunlight directions/intensities.

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
