# Peer review of "Swin-Transformer-YOLOv5 for Real-Time Wine Grape Bunch Detection"

_remotesensing, doi:10.3390/rs14225853_

Round 1
Reviewer 1 Report
Summary:
The objective the paper is to detect grape bunch in real-time, so that advantages from Swin-transformer and YoloV5 were combined in a unified architecture, replacing the last C3 block in the backbone with a Swin-transformer block. The model outperformed others, achieving mAP of more than 97% and F1-Score of 0.89 in cloudy days. The model was tested in different varieties, different weather conditions, and different sunlight conditions.
Doubts about the text:
How the Swint-T module can provide global information if it is applied as the last feature extraction block into YoloV5 backbone? Since it is employed at the end the global information will be extracted from feature maps. So I don’t agree with the affirmation “This integration can compensate for the shortcoming of YOLOv5 as one of the typical CNNs in lack of capturing global and contextual information due to the limited receptive field ” [line 284]. The text could be more clear about how Swint-T could provide global information to the model.
How the pre-trained YoloV5 model was used, since the last C3 in Yolov5 was replaced by Swin-T module into the architecture?How did you handle this difference when retraning the model. This is a important information and should be in the text.
Due to these questions "Are the methods adequately described?" was answered as "Can be improved".
More, excelent work.
Author Response
Dear Reviewer, thank you very much for your valuable time helping us improve the quality of our manuscript. We greatly appreciate your help and comments. We carefully addressed/answered all your questions and provided point-to-point responses.
Please see the attachment.
Sincerely,
Authors

Reviewer 2 Report
This paper proposed a Swin-Transformer-YOLOv5 model for real-time wine grape bunch detection. The proposed model outperformed out state-of-the-art detection models with high efficiency. A systematic study has been conducted to evaluate the model performance under different weather conditions and grape species. The paper is well written, with an adequate description of the method and appropriate interpretation of the results. But I have a few comments that I hope the authors could address or clarify before being accepted for publications. Please see my comments for more details below:
1. Line 25: what is the full term for mAP?
2. Line 54-55: It would be better to define “one-stage detection” and “two-stage detection”.
3. Line 71: “improved from 0.85 to 0.93”. What model was Improved from what model?
4. Line 280-281: The early layers of the neural networks learn general features and the bottom layers learn more task-specific features. Since the Swin-transformer is good at embedding global and local information, it makes more sense to me to use the Swin transformer the first C3 layer instead of the last C3 layer. Could you explain the reason why the Swin transformer is used in this way?
5. Line 286-289/Line 291: The input to the Swin transformer is 20-by-20 feature maps, I am not sure whether it is still worth discussing local information and long-distance dependency. The Focus module and CBL module have already embedded the global information and created dependency among features (which may not exist before).
6. Line 286-289: Not sure whether the authors did position embedding for the feature maps before being fed into the Swin transformer. If so, please give some description and discussion.
7. Line 294: Was the Swin transformer pre-trained with the COCO dataset as well?
8. Table 3: may include the number of trainable parameters for each model and training time.
9. Section 3.1: Did the training set contain images under all weather conditions? In Table 1, I noticed that image types are unbalanced as most of images are sunny. Did the authors take any steps to balance the training set?
Author Response

(The authors gave the same response as above.)

Reviewer 3 Report
The paper presents a novel object-detection architecture integrating the popular Swin Transformer and YOLOv5 models. Performance of this architecture in detecting wine grape bunches was evaluated by comparing with other commonly-used detectors. The detection accuracies were compared specifically among factors of grape varieties, environmental factors, and maturity stages.
Overall, the manuscript was very well-written with a clear organization, demonstration of importance, novelty, and a thorough discussion. A few comments can be found in the attachments for better clarification.

Author Response

(The authors gave the same response as above.)
